# Chronic Inflammatory Diseases at Secondary Sites Ensuing Urogenital or Pulmonary *Chlamydia* Infections

**DOI:** 10.3390/microorganisms8010127

**Published:** 2020-01-17

**Authors:** Yi Ying Cheok, Chalystha Yie Qin Lee, Heng Choon Cheong, Chung Yeng Looi, Won Fen Wong

**Affiliations:** 1Department of Medical Microbiology, Faculty of Medicine, University of Malaya, Kuala Lumpur 50603, Malaysia; heathercheok@gmail.com (Y.Y.C.); chalystha@gmail.com (C.Y.Q.L.); cheonghengchoon@gmail.com (H.C.C.); 2School of Biosciences, Faculty of Health and Medical Sciences, Taylor’s University, Subang Jaya 47500, Malaysia; ChungYeng.Looi@taylors.edu.my

**Keywords:** *Chlamydia trachomatis*, *Chlamydia pneumoniae*, chronic inflammation, reactive arthritis, atherosclerosis, multiple sclerosis, Alzheimer’s disease, primary biliary cirrhosis

## Abstract

*Chlamydia trachomatis* and *C. pneumoniae* are members of the *Chlamydiaceae* family of obligate intracellular bacteria. The former causes diseases predominantly at the mucosal epithelial layer of the urogenital or eye, leading to pelvic inflammatory diseases or blindness; while the latter is a major causative agent for pulmonary infection. On top of these well-described diseases at the respective primary infection sites, *Chlamydia* are notoriously known to migrate and cause pathologies at remote sites of a host. One such example is the sexually acquired reactive arthritis that often occurs at few weeks after genital *C. trachomatis* infection. *C. pneumoniae*, on the other hand, has been implicated in an extensive list of chronic inflammatory diseases which include atherosclerosis, multiple sclerosis, Alzheimer’s disease, asthma, and primary biliary cirrhosis. This review summarizes the *Chlamydia* infection associated diseases at the secondary sites of infection, and describes the potential mechanisms involved in the disease migration and pathogenesis.

## 1. Introduction

*Chlamydia* is an obligate intracellular bacterium that infects mucosal epithelial cells primarily at the urogenital, eye, and pulmonary sites. It has a unique biphasic developmental phase which alternates between elementary body (EB) or reticulate body (RB), respectively representing the extracellular and intracellular forms within its life cycle [1], as depicted in Figure 1. At EB phase, *Chlamydia* is small (0.2–0.6 μm) and compact due to densely crosslinking cysteine-rich outer membrane proteins by disulphide bonds that form a “supramolecular disulphide complex” [2]. The chlamydial EB is an osmotically stable and metabolically dormant bacterium, this permits survival at harsh extracellular environments and facilitates its attachment and entry into the host cell. After entering a cell, the EB transforms into RB that is characterized by reduced “supramolecular disulphide complex”, and appears relatively larger (0.6–1.5 μm) in size. In RB form, *Chlamydia* is osmotically fragile but metabolically active; this equips the bacteria for robust cell division through binary fission within inclusions [2,3]. The newly synthesized RBs will be converted to EBs in a process signaled by size reduction, where the RBs gradually decrease in size following multiple rounds of binary fission before differentiating into EBs [4]. Toward the final stage of the developmental cycle, the EBs are released from the host cell through extrusion or cellular lysis to commence the developmental cycle anew [5].

Suboptimal environmental stimuli such as poor nutrition, antibiotic stress, or host immune-aggression can induce *Chlamydia* to enter a persistent phase in which the RBs become aberrantly enlarged [6]. Transcriptional profiling analyses showed that these atypically large RBs are metabolically active [7,8]. This has led to the suggestion that this may be a mode of growth whereby *Chlamydia* can be protected from the host defense mechanism while stockpiling nutrients in preparation for growth when the conditions become conducive to its replication [9]. Additionally, *Chlamydia* can repurpose the host cell for its growth advantage. For instance, in human epithelial cells, it alters protein stability and proteome profile, including mammalian target of rapamycin (mTOR)-mediated pathway for energy production that facilitates RB replication in inclusion [10,11]. Strategies of immune evasion underlying chronic persistency of *Chlamydia* potentiate the pathogen’s long-term survival thus providing opportunity for bacterial dissemination from primary infectious site to a remote location [12]. As a consequence, *Chlamydia* infection-mediated pathologies extend beyond urogenital, eye, and pulmonary sites, and are associated with a gaining list of chronic inflammatory diseases, including reactive arthritis, atherosclerosis, multiple sclerosis, Alzheimer’s disease, asthma, and primary biliary cirrhosis, as summarized in Table 1. The current review focuses on the mechanisms of bacteria migration and pathogenesis of these diseases that occur at the secondary sites following *Chlamydia trachomatis* and *Chlamydia pneumoniae* infections in human host.

## 2. Chlamydial Infection at the Primary Sites

*C. trachomatis* and *C. pneumoniae* are two important human pathogens belonging to the *Chlamydiaceae* family well-known for causing urogenital, eye, and pulmonary complications. *C. trachomatis* comprises three major groups of biovariants including serovars A to C which cause trachoma, serovars D to K which cause urogenital infection, and the L1 to L3 serovars which cause lymphogranuloma venereum. Importantly, genital infection in female could lead to development of pelvic inflammatory disease (PID), ectopic pregnancy, and tubal factor infertility [61]. *C. pneumoniae* is another known member of the genus *Chlamydia* that is recognized as a human pathogen. *C. pneumoniae* is found in the majority of the world’s population as it is easily disseminated via the respiratory route. Infection with *C. pneumoniae* commonly causes acute respiratory tract infection. The *Chlamydiaceae* family includes a total of 11 other species of the genus *Chlamydia* and the taxon *Candidatus* such as *C. abortus*, *C. caviae*, *C. felis*, *C. pecorum, C. psittaci*, Ca. *C. ibidis*, among others, that infect predominantly animals including live stocks, felines, koala, and avian. Rarely, cases of zoonotic transmission of these nonhuman *Chlamydia* species to humans were reported to cause atypical pneumonia, conjunctivitis, or abortion; which resemble the diseases caused either by *C. trachomatis* or *C. pneumoniae* in humans at the primary site of infection [62]. At present, there is still no effective chlamydia vaccine available which necessitates further efforts to elucidate the intricate host-pathogen interaction.

## 3. *C. trachomatis* and Reactive Arthritis

Reactive arthritis (ReA), formerly known as Reiter’s syndrome, is an inflammatory disease that commences few weeks after an infection at a location outside of the joint. It is characterized by inflammation of the joints and tendons. The first apparent septic ReA case was recorded in France around two centuries ago when swelling of both knees was observed in individuals following sexually transmitted infections [63]. At present, compelling evidences have incriminated *C. trachomatis* (serovars D to K) as the culprit which provokes post-venereal ReA [13,14]. Infection by *C. pneumoniae* has also been associated with ReA although doubts remain concerning its role in the onset of disease [64,65], as it was detected at lower prevalence than *C. trachomatis* among ReA patients [66,67,68,69,70]. A long list of pathogens have been implicated in ReA including enteric pathogens *Salmonella*, *Shigella*, or *Campylobacter*, etc. [71]. *C. trachomatis* has been recognized as a causative agent in sexually acquired ReA using serological or molecular assays [13,14] that enable detection of the bacterial DNA, antigens, or EB in the synovial fluid or arthritic joint from patients [15,72]. A direct causative link between *C. trachomatis* and ReA development has been established in the experimental animal model of autoimmune arthritic SKG (ZAP70 W163C mutant) mouse which developed spondylo-arthritis at five weeks post-vaginal infection with murine biovar *C. muridarum* [73]. Notably, the disease severity of ReA in the SKG mice correlated with the size of bacterial inoculum, and could be prevented upon early antibiotics treatment, confirming the bacteria as a contributing factor for the progression of the disease [73].

### 3.1. Bacterial Travel Beyond Urogenital Tract, Hypoxic Environment, and Persistence Stage Fuel the Occurrence of Post-Venereal Reactive Arthritis

Several studies have proposed the mechanisms underpinning the migration of an intracellular pathogen from a primary infection to a secondary body site. *C. trachomatis* has been shown to disseminate systemically to organs including liver, lung, spleen, and synovium, via hiding in monocytic cells as their “trojan horses” [16,17]. Furthermore, *C. trachomatis* Ba and D serovars are able to persist for a long period of time and trigger a low level of Tumor necrosis factor-α (TNF-α) and Interleukin-1β (IL-1β) pro-inflammatory cytokines but a higher level of IL-10 anti-inflammatory cytokine within monocytes compared to dendritic cells; *C. trachomatis* L2 serovar is able to sustain their development cycle in both monocytes and dendritic cells [74].

Upon entering the synovium, the unique hypoxic condition in the inflamed joint creates a thriving environment for *C. trachomatis* to multiply. Interferon-γ (IFN-γ) is well-established to exert anti-chlamydial activity by activating Janus kinase-signal transducer and activator of transcription (JAK-STAT) signaling cascades that subsequently induce indoleamine 2,3-dioxygenase (IDO) [20]. Presence of IDO limits chlamydial growth by promoting degradation of tryptophan, an amino acid essential for chlamydial replication. Hypoxic environment in the synovium abrogates the antimicrobial activity of IFN-γ, accompanied by reduced STAT1 phosphorylation and diminished IDO activity [18,19]. Nutrient deprivation condition following IDO-mediated degradation of tryptophan also promotes *C. trachomatis* to escape from a conventional EB-RB cycle, and enter a persistence stage [20]. This ability, also seen in *Waddlia chondrophila*, a close relative of *C. trachomatis*, allows long term survival within the host cells in times of stress [75]. It is important to note that IFN-γ knockout but not IDO knockout mouse demonstrated a significantly higher cervico-vaginal shedding of *C. trachomatis* serovar D at two weeks post bacteria inoculation, this implies involvement of other interferon-related genes in addition to IDO [76].

### 3.2. Molecular Mimicry and Inflammatory Property of Chlamydial Antigens Triggers Local Inflammatory Response in Reactive Arthritis

As with some other autoimmune diseases, molecular mimicry of the human antigen with chlamydial antigen has been suggested as one of the major contributing factors for the onset of ReA. It has been reported that the chlamydial HSP60 is highly identical to the human HSP60; thus, an infection could lead to the production of autoantibodies that cross-react with human HSP60 [40]. HSP60-specific T cell clone is detected among many other chlamydial specific T cells that were isolated from the synovial fluid of the sexually acquired ReA [21,22].

In addition to the antigens causing molecular mimicry, other chlamydial proteins such as major outer membrane protein (MOMP) and chlamydial protease-like factor (CPAF) are able to elicit production of pro-inflammatory cytokines (TNF-α, IL-1β, IL-6) from human peripheral blood mononuclear cells [77]. For instance, MOMP induces suppressor of cytokine signaling (SOCS3) which affects the polarization of macrophage to type 1 macrophage phenotype (M1) that is inflammatory [78]. Infection of SKG mice with a high chlamydial burden elevates TNF-α secretion, but not IFN-γ and IL-17A in the SKG experiment model [73]. Hence, autoimmune and inflammatory reactions subsequent to chlamydial infection could lead to uncontrolled immune responses that exacerbates disease severity in ReA patients.

### 3.3. Antibiotic Treatment for Reactive Arthritis

Understanding the causal linkage of ReA and *C. trachomatis* enables therapeutic option with antibiotics prescription. Combination therapy with doxycycline or azithromycin and rifampin has been shown to eliminate the bacteria and improve clinical outcomes of the patients [79,80]. Nonsteroidal anti-inflammatory drugs (NSAIDs), corticosteroids, and disease-modifying anti-rheumatic drug (DMARD) such as sulfasalazine are primary choices for effective treatment of ReA [81]. As an active transcription of TNF-α has been reported in the T cells derived from ReA patients [82], TNF-α inhibitor which suppresses inflammatory response is also being developed as a potential choice of treatment for ReA [83]. Despite this, a meta-analysis of overall clinical studies reported the equivocal efficacy of antibiotics on ReA [84]. Therefore, more relevant studies should be anticipated to look into the chlamydial modulation and evasion of human innate and adaptive immunity, which enables its persistence, travel, and disease at the joint or other secondary infection sites.

## 4. *C. pneumoniae* and Atherosclerosis

Atherosclerosis is a progressive inflammatory disease caused by the hardening and narrowing of blood vessels that limit the nutrient and oxygen supplies to organs. Pulmonary infection with *C. pneumoniae* has been associated with development of atherosclerosis [85,86,87]. Many studies of recent years have shown that *C. pneumoniae* can be found in atherosclerotic plaques [23], and its presence exacerbates atherosclerosis pathology [24,25]. The first suggestion of a possible association between *C. pneumoniae* and atherosclerotic cardiovascular diseases came from a study by Saikku et al. [88], who discovered that the patients with acute myocardial infarction and chronic coronary heart disease had significantly higher anti-*Chlamydia* antibody titres as compared with healthy controls. This finding in 1988 soon snowballed into many succeeding studies focusing on the role of *C. pneumoniae* in the pathogenesis, progression, and acceleration of atherosclerosis. A recent study on 115 Moroccan patients with cardiovascular diseases (CVD) showed high prevalence of *C. pneumoniae* within the population [23]. *C. pneumoniae* DNA was detected in both patient peripheral blood mononuclear cell and atheroma plaque samples at prevalence rates of 61% and 86%, respectively [23]. Notably, 12% of the study population presented only *C. pneumoniae* infection as a sole risk factor for CVD, where other risk factors accounted for include tobacco and alcohol use, diabetes, and high blood pressure [23]. This finding contributes to the hypothesis that *C. pneumoniae* infection may play a causative role in atherosclerosis that precedes CVD.

### 4.1. Adhesive C. pneumoniae-Infected Monocytes and Foam Cell to Endothelium Enhances Atherosclerotic Plaque Formation

It has been speculated that the *C. pneumoniae* infection may affect the onset of inflammatory lesion and the rupture of atherosclerotic plaques, but the exact mechanism of how and at which stage it is involved in is still under investigation. Using imaging flow cytometry, Evani et al. demonstrated that the rolling of infected monocytes on E-selectin and endothelial surfaces was slower and more uniform, which translates to a firmer adhesion of the monocyte to the endothelium [26]. In mice kept on a normal and nonhigh fat diet, infection with *C. pneumoniae* resulted in an increase in aorta and blood cholesterol levels, along with an upregulated expression of cholesterol transporters [89].

One of the key features of the pathogenesis of atherosclerosis is inflammatory foam cell formation. Foam cells are macrophages that accumulate lipid due to decreased outflow or increased inflow and esterification of cholesterol, and are commonly found in atherosclerotic plaques to cause continuous inflammatory stimulus [90]. *C. pneumoniae* infection accelerates atherosclerosis by promoting formation of foam cells [24]. *C. pneumoniae* induces activation of NOD-like receptor protein 3 (NLRP3) inflammasome which triggers interleukin-1β (IL-1β) secretion, and in turn suppresses cholesterol efflux, leading to accumulation of cholesterol within the cell and enhancing foam cell formation [24].

### 4.2. TLR4- and MyD88-Dependent Host Immune Response Accelerates Chlamydial-Mediated Atherosclerosis

Several studies have suggested that certain features of the immune system may aid in the development of *C. pneumoniae*-induced atherosclerosis. Chen et al. reported that while atherosclerosis was significantly accelerated in *C. pneumoniae*-infected TLR4- and MyD88-positive mice, *C. pneumoniae* infection resulted in minimal effect on atherosclerosis progression in TLR4- and MyD88-deficient mice [25], indicating the importance of the TLR4/MyD88 pathway in *C. pneumoniae*-mediated atherosclerosis. Another study used *CD8* knockout mice to demonstrate that CD8^+^ T cells play a significant role in *C. pneumoniae*-induced development of atherosclerosis [27]. The infected *CD8* knockout mice had markedly lesser atherosclerotic plaque lesions as compared to wildtype mice. Repletion of *CD8* knockout mice with wildtype CD8^+^ T cells at the point of infection resulted in lesions comparable to wildtype levels.

### 4.3. Platelet Aggregation by C. pneumoniae Enhances Atherosclerotic Plaque Formation

*C. pneumoniae* infection can also indirectly increase the risk factors that lead to atherosclerotic lesions. Kälvegren et al. discovered that *C. pneumoniae* was highly adhesive to platelets and triggered its activation and aggregation [29]. Increased platelet aggregation can lead to thrombus formation and hence increased risk of atherosclerosis. The same group also reported that incubation of platelets with *C. pneumoniae* led to the production of reactive oxygen species (ROS) and the oxidation of low-density lipoprotein (LDL) to oxLDL [28]. Therefore, *C. pneumoniae* is associated with platelet activation that contributes to the initiation and development of atherosclerosis.

## 5. *C. pneumoniae* and Multiple Sclerosis

Multiple sclerosis (MS) is a progressive demyelinating and inflammatory disease of the central nervous system characterized by a large focal lesion in the white matter of the brain and spinal cord with gliosis around the blood vessels [91]. As with many other autoimmune disorders, the aetiological agent for MS disease is unknown, but it is often attributed to a combination of genetic and environmental influences. An association of MS with infection has been previously proposed as a high IgG and oligoclonal bands are detected in the cerebrospinal fluid (CSF) from the MS patients [92]. Pathogens implicated include *C. pneumoniae*, *Staphylococcus aureus*, Herpesviridae viruses, and human endogenous retrovirus.

The association of *C. pneumoniae* with MS progression was first suggested in 1998 by Sriram et al. who reported the presence of *C. pneumoniae* in the blood and CSF from a 24-year-old man with rapidly progressive MS [30]. Study using a culture method demonstrated that patients with either relapsing-remitting or progressive MS has a higher percentage (64%) of positive infection than patients with other neurological disorders (11%). Such high percentages of infection were also identified using PCR amplification of OmpA gene (97%) and antibodies titres (86%) [31], suggesting a strong correlation between the infection and MS progression [32,33,34]. A meta-analysis showed that CSF levels of *C. pneumoniae* DNA by PCR and intrathecally synthesized immunoglobulins are more likely to be detected in MS patients than patients with other neurological disorders or healthy controls [93].

However, conflicting results have also been obtained, which may be due in large part to incoherent detection methods [94]. Hammerschlag et al. (2000) has failed to observe a similar trend in their study using PCR detection and culture identification in three different laboratories [35]. It has also been argued that the presence of *C. pneumoniae* DNA in CSF is not restricted to MS, but rather a common occurrence in other brain diseases. Supporting this, Gieffers et al. found that 21% MS patients and 43% patients with other neurological disorders had *C. pneumoniae* DNA in the CSF [36]. Reports on nonspecific detection of intrathecal antibody production against the organism lends further support the association of *C. pneumoniae* with MS [95]. Nonetheless, a definitive correlation between *C. pneumoniae* and the development of MS is yet to be established and hence warrants further investigations.

### 5.1. C. pneumoniae Infects Brain Endothelial Cells and Disrupts Integrity of Blood-Brain Barrier

A few hypotheses have been put forth on the pathological involvement of *C. pneumoniae* in MS. Firstly, *C. pneumoniae* is able to infect endothelial cells in the blood vessels [96] and human brain microvascular endothelial cells [37]. Infection with *C. pneumoniae* reduces expression of Occludin protein thus disrupting the tight junctions of blood-brain barrier (BBB) [37]; this enables transmigration of leukocytes into the brain [38]. In addition, in vitro study demonstrated the ability of *C. pneumoniae* to infect astrocytes and microglia [39]. Hence, the bacteria can disseminate to the brain via monocytes or even direct entry in EB form with the impairment in the BBB, resulting in enhanced neuro-inflammation through induction of neuronal cell death. Genetic factors may also play a role in increasing the susceptibility. For example, Apolipoprotein E (APOE4) risk allele enhances the attachment of chlamydial EB to the host cells [97], while absence of a susceptible genetic make-up leads to incomplete form of the MS disease [98].

### 5.2. Molecular Mimicry

Another form of pathogenesis in MS includes molecular mimicry of the HSP60 that leads to production of autoantibodies that cross-reacts with human HSP60, and causes inflammation, as reviewed above [40]. On top of that, a *C. pneumoniae* 20-mer peptide has been reported to be identical to a 7-amino acid motif with the critical epitope of myelin basic protein. This peptide is able to induce T helper 1 (T_H_1) response in experimental autoimmune encephalomyelitis within infected Lewis rats, which closely resemble the pathology of MS [41]. Despite these reports, long term antibiotic treatment with roxithromycin on MS patients has failed to observe any significant benefits, suggesting that the pathology of the disease is probably more complex than originally thought [99,100].

## 6. *C. pneumoniae* and Alzheimer’s Disease

Alzheimer’s Disease (AD) is a form of progressive, irreversible neurodegenerative disease that accounts for 60% to 80% cases of dementia [101]. Neuropathological hallmarks of AD include the deposition of Amyloid-β (Aβ) plaques derived from Amyloid Precursor Protein (APP) and neurofibrillary tangles (NFT) made up of the Tau protein, leading to progressive cognitive and memory impairment accompanied by alteration in behaviour and personality [102,103].

The association of *C. pneumoniae* with late-onset AD has remained controversial after being first reported in year 1998. Balin et al. reported the presence of live and metabolically active *C. pneumoniae* in areas of pathology [42]. In this study, the presence of chlamydial DNA was detected by PCR and RT-PCR; EB and RB was also shown using immunohistochemistry (IHC) and electron microscopy. The finding was later supported by a correlation study [44] and IHC investigation [43]. Findings from several other investigations, however, disagree with an importance of *C. pneumoniae* in AD. Failure of bacterial detection in AD samples were reported using paraffin embedded tissue sample [47], frozen brain section [48], as well as PCR amplification [49]. Using animal models, Little et al. (2004) demonstrated that BALB/c mice intranasally infected with the *C. pneumoniae* (96–41 strain) developed Aβ deposits in their brains [45,46]. Nevertheless, Boelen et al. (2007) failed to reproduce the result using a different bacterial strain (Taiwan acute respiratory agent) and observed no significant difference between the mock- and *C. pneumoniae*-infected mice [104]. To date, the involvement of *C. pneumoniae* in AD still remain debatable due to discrepancies of findings in both human and animal studies.

### 6.1. C. pneumoniae Infected Monocytes Gain Penetration through Brain Endothelial Cells

Various hypotheses have been proposed to help explain the pathogenesis of the bacteria in late-onset AD. As described above, *Chlamydia* is able to disseminate to the vasculature through hiding in monocytes [16,17] and spread to the brain when these monocytes penetrate the blood brain barrier and enter the brain. In a report by MacIntyre et al. (2003), *C. pneumoniae* has been identified to be able to promote such transmigration of infected monocytes into the brain via human brain endothelial cells [50]. On the other hand, it has been described that the attachment of chlamydial EB to the host cells is mediated by APOE4 [97], supporting previous finding which shows higher bacterial load of *C. pneumoniae* in Alzheimer’s patients carrying this risk allele [105].

### 6.2. Inflammatory LPS-Induced Activation of Microglial

Shedding of the highly inflammagenic lipopolysaccharide (LPS) from cell wall of Gram-negative bacteria is perhaps the most widely discussed pathogenic mechanism of AD. AD patients’ blood has 3-fold higher level of LPS compared to healthy control [106]. LPS is able to induce inflammation through activation of cytokines tumor necrosis factor-α (TNF-α), interleukin-1β (IL-1β), and IL-6 production and recruitment of microglial or astrocytes to the affected area. Such neuropathogenesis may later on affect gene transcription involved in the formation of Aβ plaques [51]. The amyloid structure could be reversed by the addition of LPS binding protein (LBP) which prevents LPS-induced blood clot into amyloid form [107].

LPS induction in mouse model induces inflammation through activating nuclear factor kappa B (NF-κB) signaling in microglial cells, leading to elevated levels of TNF-α, IL-1β, prostaglandin E2 (PGE2), and nitric oxide (NO), and consequently causes neuroinflammation and cognitive impairment [108]. Intraperitoneal injection of LPS resulted in an accumulation of Aβ1–42 in the hippocampus and cerebral cortex of mice brains that can be alleviated through a pre-treatment of oral anti-inflammatory agent [109].

### 6.3. Antibiotic Treatment for Alzheimer’s Disease

Antibiotic therapy has been evaluated as an intervention strategy for the treatment of AD. The first randomized-control trial on the effectiveness of doxycycline and rifampicin on AD patients was carried out in 2004. These antibiotics have successfully reduced cognitive worsening in late onset AD patients over the course of six months. However, it is indicated that there was no significant reduction in the *C. pneumoniae* IgA and IgG antibody titres, and a low prevalence of chlamydial DNA in the blood (6%) has also restricted the attempt to establish their correlation [110]. In addition, rifampicin has a validated therapeutic efficacy in preventing Aβ and Tau oligomers deposition in both in vitro and in vivo experiments [111,112,113]. Therefore, the decrement in cognitive worsening may be attributed to the effects of the antibiotics alone rather than the clearance of infection.

## 7. *C. pneumoniae* and Asthma

Asthma is an inflammatory lung disease that is characterized by airway hyperresponsiveness towards various triggers of bronchoconstriction. An increasing amount of evidences indicate that certain members of the respiratory tract microbiome may underlie the development of this disease. Among the plethora of microbial colonizers inhabiting the airway, *C. pneumoniae* is frequently incriminated as a possible infectious aetiological agent of asthma attacks [54,114,115,116]. In addition to initiating symptoms of asthma, *C. pneumoniae* was also found to be associated with disease severity [52,53,117]. Hahn et al. demonstrated that *C. pneumoniae*-specific IgE was detected in 79% of patients in a study cohort of 66 severe and persistent asthma patients [53]. Cook et al. discovered the presence of *C. pneumoniae* markers in asthmatic patients that have symptoms that are more severe than the norm and more difficult to control [118]. Cunningham et al. hypothesized that the host immune response to chronic infection of *C. pneumoniae* exacerbates asthma symptoms by interacting with host allergic inflammation [52]. Emerging evidence also points to the association between *C. pneumoniae* infection and steroid resistance in asthma patients [119]. One of the known factors that causes steroid resistance in asthma patients is due to the inhalation of cigarette smoke [120]. *C. pneumoniae* infection also induces pulmonary bronchial epithelial ciliostasis, a condition similarly observed in asthma patients exposed to cigarette smoke [121]. Immune cells and smooth muscle cells found throughout the respiratory tract that is infected with *C. pneumoniae* results in an increased secretion of inflammatory cytokines and chemokines [55]. This contributes to the onset of hyperreactivity in the bronchial airways, a characteristic feature of asthma.

## 8. *C. pneumoniae* and Primary Biliary Cirrhosis

Primary biliary cirrhosis (PBC) is a chronic cholestatic inflammatory disease characterized by immune-mediated hepatic portal inflammation and destruction of the bile ducts that cause liver failure [122]. Up to 95% of PBC patients produce a high titre of serological anti-mitochondrial antibodies (AMA) [123]. At present, the aetiology of PBC is unknown. It has been speculated that microbial infection triggers the onset of the disease by molecular mimicry between bacteria and host antigens or direct presence of the bacteria at the liver [60]. A study carried out by Abdulkarim et al. has brought *C. pneumoniae* infection to attention as a candidate that triggers PBC [56]. This group reported that *C. pneumoniae* antigens was detected in the explanted liver tissue or liver biopsy of all patients in PBC group, compared to only 8.5% in the group of other chronic liver diseases. In addition, *C. pneumoniae* 16S rRNA was also detected in the PBC patient by in situ hybridization. In contrast to this study, most other studies do not support a direct association between *Chlamydia* and PBC. A study led by Leung et al. investigated the serological activity of *C. pneumoniae* and *C. trachomatis* in PBC patients, as well as the presence of the bacteria in liver tissue samples [57]. The outcome of this study showed that although the antibody response to *Chlamydia* antigens was significantly higher in AMA-positive PBC patients as compared to AMA-negative PBC patients (91% versus 21%), the group was unable to detect the presence of *Chlamydia* in the PBC liver tissue samples. Liu et al. reported that *C. pneumoniae*-specific IgG levels were significantly higher in PBC patients than healthy controls, but showed no difference when compared to pathological controls with post-hepatic cirrhosis [58]. A recent study by Arcos et al. showed no significance in serum antibodies against *C. pneumoniae* and bacterial DNA was not detected in the liver tissues of the patients [59]. As such, there is currently insufficient evidence to state that PBC is directly associated with *Chlamydia* infection.

## 9. Conclusions

Long term persistence of *Chlamydia* in the host contributes to chronic inflammatory diseases at remote sites of the body through various mechanisms; either through bacterial persistence, migration and infection at the secondary site, or indirectly by triggering immune response through molecular mimicry and secretion of bacterial products which activate inflammatory mediators. It is noteworthy that the inflammatory diseases discussed above are multifactorial and a pathogenic agent may not be the sole element responsible for the disease in the host. Attention should also be given to other factors including genetic susceptibility, host immune response, as well as the environment such as geography, climate, etc. Last but not least, the causal link between *Chlamydia* and the chronic inflammatory diseases calls for more investigations to elucidate the disease mechanism, which may provide a better clue for appropriate disease management.

## Figures and Tables

**Figure 1 microorganisms-08-00127-f001:**
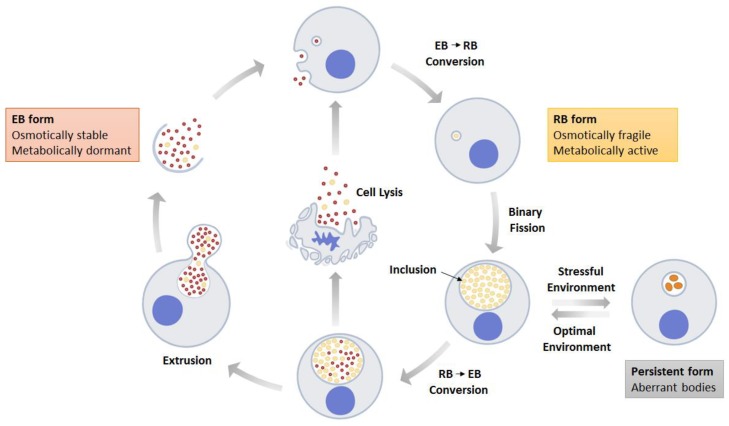
Schematic diagram of the developmental cycle of *Chlamydia*. Elementary bodies (EB; red dots) undergo conversion to reticulate bodies (RB; yellow dots) following attachment and internalization into the host cell. RB multiplies by binary fission and matures into EB before being released by lysis or extrusion processes. In the presence of growth stress such as IFN-γ, the RB enlarges and enters persistence. Optimal condition causes the enlarged RB to return to normal growth.

**Table 1 microorganisms-08-00127-t001:** The list of chronic inflammatory diseases associated with infection by the *Chlamydiaceae* family.

Diseases	Supporting or Contradicting Evidence	Mechanisms of Pathogenesis
Reactive Arthritis (ReA)	*Supporting:* Presence of *C. trachomatis* DNA, antigens, EB in the synovial fluid; elevated serum anti-*C. trachomatis* antibodies in ReA patients [13,14,15].	i. *Chlamydia* hijacks monocytic cells as their “trojan horses” [16,17] to travel to synovium where hypoxic stress inhibits indoleamine 2,3-dioxygenase (IDO) activity [18,19] and nutrient starvation promotes bacteria persistency [20]. i. Molecular mimicry between host and chlamydial HSP60 proteins [21,22] and presence of other *Chlamydia* antigens triggers robust secretion of inflammatory cytokines.
Atherosclerosis	*Supporting:* Presence of *C. pneumoniae* in atherosclerotic plaques exacerbates disease pathology; elevated anti-*C. pneumoniae* antibodies among patients [23,24,25].	i. *Chlamydia* facilitates plaque formation by enhancing a firm adhesion of the monocyte to the endothelium [26] and promotes foam cells formation [24]. i. *Chlamydia* accelerates development of atherosclerosis by activating TLR4 signaling pathway [25], and CD8+ T cells [27]. i. *C. pneumoniae* adheres to platelets and causes aggregation that increases risk of atherosclerosis [28,29].
Multiple Sclerosis (MS)	*Supporting:* High percentage of positive *C. pneumoniae* infection using culture method; positive amplification of chlamydial OmpA gene [30,31]; presence of anti-*C. pneumoniae* antibodies in MS patients [31,32,33,34]. *Contradicting:* Failure to detect bacteria in MS patients using PCR or culture methods [35]; presence of bacteria in other neurological diseases in addition to MS [36].	i. *Chlamydia* disrupts blood-brain barrier (BBB) [37] and enables bacteria dissemination through monocyte or EB transmigration into brain [38] where it causes neuroinflammatory lesion by infecting astrocytes and microglia [39]. i. Molecular mimicry of HSP60 and a bacteria peptide that mimics human myelin basic protein leads to production of cross-reactive autoantibodies causing inflammation [40,41].
Alzheimer’s Disease	*Supporting:* Presence of live and metabolically active *C. pneumoniae* in brain of Alzheimer’s disease patients [42,43,44]; intranasal *C. pneumoniae*-infected mice developed brain Aβ deposits [45,46]. *Contradicting:* Failure of bacterial detection in patients’ brain section [36,47,48], or PCR amplification [49].	i. *Chlamydia* disseminates to brain through hiding in monocytes and disrupting junction at the human brain microvascular endothelial cells [16,17,37,50]. i. *Chlamydia* shedding of the lipopolysaccharide (LPS) activates nuclear factor kappa B (NF-κB) and promotes inflammatory cytokines production [51].
Asthma	*Supporting: C. pneumoniae*-specific IgA, IgE, IgG, and IgM antibodies were detected in asthmatic patients [52,53,54]; IgE showed a positive linear association with asthma severity [53]. Children who reported multiple episodes of asthma were PCR positive for *C. pneumoniae* [52].	*C. pneumoniae* contributes to asthma by causing increased secretion of inflammatory cytokines and chemokines [52,55].
Primary Biliary Cirrhosis (PBC)	*Supporting: C. pneumoniae* antigens and 16S rRNA was detected in the explanted liver tissue or biopsy [56]; elevated anti-chlamydial antibodies among PBC or anti-mitochondrial antibodies (AMA)-positive PBC patients [57,58]. *Contradicting:* Absence of bacteria or bacterial DNA in the liver tissues; no significant serum anti-*C. pneumoniae* IgG in PBC patients [57,58,59].	Potentially due to molecular mimicry between bacteria and host antigens or direct presence of the bacteria at liver [60], however an exact mechanism is still unknown.

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
