# Peer review of "Chronic Inflammatory Diseases at Secondary Sites Ensuing Urogenital or Pulmonary Chlamydia Infections"

_microorganisms, 2020, doi:10.3390/microorganisms8010127_

Round 1

Reviewer 1 Report

Very well done! This manuscript is all about Chronic Inflammation, a subject very close to my heart, and its implications.

Author Response

We thank the reviewer for generous comments given.

Reviewer 2 Report

The manuscript “Chronic inflammatory diseases at secondary sites ensuing urogenital or pulmonary Chlamydia infections” by Yi Ying Cheok, Chalystha Yie Qin Lee, Heng Choon Cheong, Chung Yeng Looi, and Won Fen Wong, faces the relevant and growing topic of Chlamydia infection and the  Chlamydia-associated chronic inflammatory diseases located in distant areas from infection site. The manuscript is generally well written and updated, even though the pre-fixed targets (mechanism of Chlamydia migration from infection site to distant areas and pathogenesis of the observed pathologies in these distant areas) appear quite limited and lacking of a relevant topic, as the diagnosis of infection.

However, in order to make clearer the content for the readers, the following points should be addressed:

1.     It is helpful to add a figure in the Introduction, reporting the life-cycle of Chlamidia

2.     At pag. 2, line 22: L serovars” should be singular (“L serovar”); moreover, in the same paragraph, a table reporting the organization of Chlamydiaceae family, like the one reported in the reference 14, should be useful for the readers

3.     The paragraphs dedicated to the Chlamydia-associated pathologies in distant sites should be separated and different from the paragraph 2 on the Chlamydial infection at the primary sites, thus the paragraph on reactive arthritis should be 3 instead than 2.

4.     In my opinion reactive arthritis and atherosclerosis deserve a higher emphasis than multiple sclerosis, Alzheimer’s disease, asthma and primary biliary cirrosis which may only be mentioned.

By now, indeed, the first are demonstrated distant sites pathologies whereas the latter present scarce and contrasting results in the literature. Treating all these pathologies with the same emphasis may result misleading for the readers.

5.     The statement at pag. 4, line 1 is incorrect. Is well established that NSAIDs are first-line drugs for the management of reactive arthritis, refractory disease can be treated with nonbiological disease-modifying agents such as sulfasalazine. The role of tumor necrosis factor alpha inhibitors is developing.

Minor points:

·      STI (page 2 line 40) should be reported in extenso as “sexually trasmitted infections” and, if no more used, without the abbreviation in STI

·      Pag 4, lines 1-2, “a meta analyses” should be “a meta analysis”;

·      Page 5, line 14: “(CNS)” if no more used it is useless to abbreviate; line 18: CSF should be put in extenso here and not in the line 22

·      Page 6, lines 6-7: “heat shock protein 60 (HSP-60)” has been already reported in extenso at page 3, line 32

·      Page 6, line 22: “by” seems to be useless and should be eliminated.

Author Response

We thank the reviewer for kind comments given.

The manuscript “Chronic inflammatory diseases at secondary sites ensuing urogenital or pulmonary Chlamydia infections” by Yi Ying Cheok, Chalystha Yie Qin Lee, Heng Choon Cheong, Chung Yeng Looi, and Won Fen Wong, faces the relevant and growing topic of Chlamydia infection and the  Chlamydia-associated chronic inflammatory diseases located in distant areas from infection site. The manuscript is generally well written and updated, even though the pre-fixed targets (mechanism of Chlamydia migration from infection site to distant areas and pathogenesis of the observed pathologies in these distant areas) appear quite limited and lacking of a relevant topic, as the diagnosis of infection.

However, in order to make clearer the content for the readers, the following points should be addressed:

It is helpful to add a figure in the Introduction, reporting the life-cycle of Chlamydia

> A Figure to describe Chlamydia life-cycle has been added (please see Figure 1).

At pag. 2, line 22: L serovars” should be singular (“L serovar”); moreover, in the same paragraph, a table reporting the organization of Chlamydiaceae family, like the one reported in the reference 14, should be useful for the readers

> Page 3 Line 6: We retained the word “serovar” as plural; but changed “L serovars” to “L1 to L3 serovars”. We added a Table to summarize the chronic inflammatory diseases caused by Chlamydiaceae family, as suggested (Please see Table 1).

The paragraphs dedicated to the Chlamydia-associated pathologies in distant sites should be separated and different from the paragraph 2 on the Chlamydial infection at the primary sites, thus the paragraph on reactive arthritis should be 3 instead than 2.

> We have changed paragraph on Reactive arthritis as “3” instead of “2”, as suggested.

In my opinion reactive arthritis and atherosclerosis deserve a higher emphasis than multiple sclerosis, Alzheimer’s disease, asthma and primary biliary cirrosis which may only be mentioned.

By now, indeed, the first are demonstrated distant sites pathologies whereas the latter present scarce and contrasting results in the literature. Treating all these pathologies with the same emphasis may result misleading for the readers.

> We agreed with the reviewer, as such we have emphasized both supporting and contradicting evidences in the Table 1.

The statement at pag. 4, line 1 is incorrect. Is well established that NSAIDs are first-line drugs for the management of reactive arthritis, refractory disease can be treated with nonbiological disease-modifying agents such as sulfasalazine. The role of tumor necrosis factor alpha inhibitors is developing.

> Page 4 Line 37: We have revised the sentence and added references. “Nonsteroidal anti-inflammatory drugs (NSAIDs), corticosteroids and disease-modifying anti-rheumatic drug (DMARD) such as sulfasalazine are primary choices for effective treatment of ReA [81]. As active transcription of TNF-α has been reported in the T cells derived from ReA patients [82], TNF-α inhibitor which suppresses inflammatory response is also being developed as a potential choice of treatment for ReA [83].”

Minor points:

STI (page 2 line 40) should be reported in extenso as “sexually trasmitted infections” and, if no more used, without the abbreviation in STI Pag 4, lines 1-2, “a meta analyses” should be “a meta analysis”; Page 5, line 14: “(CNS)” if no more used it is useless to abbreviate; line 18: CSF should be put in extenso here and not in the line 22 Page 6, lines 6-7: “heat shock protein 60 (HSP-60)” has been already reported in extenso at page 3, line 32 Page 6, line 22: “by” seems to be useless and should be eliminated.

> We have corrected the minor points raised by the reviewer, as below:

Page 3 Line 25: Extenso for “STI” has been removed.

Page 4 Line 41: “Meta analyses” has been changed to “meta analysis”

Page 6 Line 5: Extenso for “CNS” has been removed, and extenso for “CSF” is mentioned in this paragraph.

Page 6 Line 45: Repeated Extenso for “HSP60” has been removed.

Page 7Line 12: The word “by” has been removed.

Round 2

Reviewer 2 Report

The Author's exhaustively answered to my observations. I have no further corrections.